# Realizing Broadband NIR Photodetection and Ultrahigh Responsivity with Ternary Blend Organic Photodetector

**DOI:** 10.3390/nano12081378

**Published:** 2022-04-18

**Authors:** Yang-Yen Yu, Yan-Cheng Peng, Yu-Cheng Chiu, Song-Jhe Liu, Chih-Ping Chen

**Affiliations:** 1Department of Materials Engineering, Ming Chi University of Technology, New Taipei City 243, Taiwan; m09188024@o365.mcut.edu.tw; 2Department of Chemical Engineering, National Taiwan University of Science and Technology, Taipei City 106335, Taiwan; ycchiu@mail.ntust.edu.tw; 3Taiwan Thompson Painting Equipment Co., Ltd., New Taipei City 25169, Taiwan; johnny@thompsontw.com

**Keywords:** organic photodetector, high performance, fullerene, responsivity

## Abstract

With the advancement of portable optoelectronics, organic semiconductors have been attracting attention for their use in the sensing of white and near-infrared light. Ideally, an organic photodiode (OPD) should simultaneously display high responsivity and a high response frequency. In this study we used a ternary blend strategy to prepare PM6: BTP-eC9: PCBM–based OPDs with a broad bandwidth (350–950 nm), ultrahigh responsivity, and a high response frequency. We monitored the dark currents of the OPDs prepared at various PC_71_BM blend ratios and evaluated their blend film morphologies using optical microscopy, atomic force microscopy, and grazing-incidence wide-angle X-ray scattering. Optimization of the morphology and energy level alignment of the blend films resulted in the OPD prepared with a PM6:BTP-eC9:PC_71_BM ternary blend weight ratio of 1:1.2:0.5 displaying an extremely low dark current (3.27 × 10^−9^ A cm^−2^) under reverse bias at −1 V, with an ultrahigh cut-off frequency (610 kHz, at 530 nm), high responsivity (0.59 A W^–1^, at −1.5 V), and high detectivity (1.10 × 10^13^ Jones, under a reverse bias of −1 V at 860 nm). Furthermore, the rise and fall times of this OPD were rapid (114 and 110 ns), respectively.

## 1. Introduction

Solution-processed organic photodetectors (OPDs) have the potential to replace inorganic photodetectors because of their light weight, flexibility, potentially lower cost (e.g., combining solution processing with roll-to-roll manufacture), and ability to select wavebands by using organic semiconductors with various chemical structures [1,2,3,4]. Accordingly, OPDs could become key components in modern technologies based on light-to-electrical signals—not only in photodetectors but also in, for example, high-speed data transmission, medical imaging, and environmental lighting [5,6]. There are many examples of light-sensing materials that operate in various wavelength bands, including the near-infrared (NIR), visible, X-ray, and ultraviolet [5,6,7,8,9,10,11,12,13]. Furthermore, there have been many recent studies aimed at improving the responsivity (*R*) or decreasing the dark current (*J*_d_) of OPDs, as well as determining the relationship between the two [6,13,14]. At present, the linear dynamic range (LDR) of OPDs can compete with that of inorganic photodetectors [2,15,16], but simultaneously exhibiting high responsivity, a fast response frequency, and an ultralow dark current (*J*_d_) at reverse bias remains a challenge for solution-processed OPDs.

Rapid progress in nonfullerene acceptors (NFA) has led to the power conversion efficiency (PCE) of organic photovoltaics (OPVs) exceeding 18% [17]. The development of NFAs had also benefitted the preparation of OPDs incorporating these state-of-the-art OPV materials. Appendix A summarizes the performance of recently reported OPDs. For example, Kim and co-workers prepared an indigo-based NFA:P3HT OPD operating with an absorption wavelength of 350–650 nm and with a value of *J*_d_ of 2.9 × 10^−8^ A cm^−2^ (at −3 V) and a high detection ratio (*D**) of approximately 10^12^ Jones [18]. Brabec et al. reported an IDTBR:P3HT OPD that provided a value of *J*_d_ of 2 × 10^−8^ A cm^−2^ (at −5 V) and a high responsivity (*R*) of 0.42 A W^−1^ in the NIR region [10]. Wang and co-workers developed an indigo-based PBDTTT-EFT: eh-IDTBR OPD exhibiting a value of *J*_d_ of 1.13 × 10^−9^ A cm^−2^ and a high value of *D** of 1.63 × 10^13^ Jones at −1 V (notably, this value of *D** is the highest reported in the literature) [2]. Gasparini et al. described a visible-light-sensitive OPD (prepared using a blend of PTQ10:O-FBR or O-IDTBR) showing an ultralow value of *J*_d_ of 1.7 × 10^−10^ A cm^−2^ (at −2 V) and a high response speed (cut-off frequency: 110 kHz) [19]. Zhang et al. prepared a double-layer (PM6:Y6/P3HT:PC_71_BM) OPD displaying an LDR of 158 dB at −5 V [20]. Although these studies demonstrate the great advancements that have been made in the state-of-the-art solution-processed OPDs, there remains plenty of room to realize high-performance OPDs with simultaneously optimized responsivities, detection ratios, and response speeds.

Ternary blend strategies can greatly improve the performance of OPDs and lead to great success in OPV applications [21]. The third component in a ternary blend can play several roles: modifying the morphology, altering the energy levels, and extending the range of light absorption. OPDs based on ternary blends have also displayed improved device performance [22,23,24,25]. OPDs and OPVs operate at reverse and forward biases, respectively. A thicker blend film is required for an OPD and, indeed, its optimized morphology can vary between applications. For example, high crystallinity of the blend moieties can lead to efficient carrier transport and improve the performance of OPVs under balanced electron/hole extraction. In OPDs, such high crystallinity for the donor or acceptor domain can contribute to the efficient injection of carriers under reverse bias [19,26,27,28,29,30]. In this paper, we demonstrate PM6: BTP-eC9–based OPDs prepared by incorporating various concentrations of (6,6)-phenyl-C_71_-butyric acid methyl ester (PC_71_BM) in the blend film. We evaluated the optoelectronic properties of these OPDs and the relationship between their performance and the blend film morphology. In general, the intrinsic *J*_d_ of an OPD can be attributed to charge carrier injection from the electrodes under reverse bias or to the thermal generation of carriers within the active layer [6]. The energy levels of the electrodes, transporting layers, and active layer materials govern the values of *J*_d_ of OPDs. We observed that efficient electron traps and a deeper highest occupied molecular orbital (HOMO) of PC_71_BM contributed to the ultralow values of *J*_d_ of our OPDs, by decreasing the excessive injection current [5]. Furthermore, the presence of PC_71_BM tailored the hole and electron mobilities of the active layer, allowing efficient carrier recombination with short rise and fall times [2,19,24,29,31]. Our optimized OPD displayed an ultrahigh value of *R* of 0.59 A W^−1^ in the IR region with a value of *D** of greater than 10^12^ Jones and an impressive cut-off frequency of 0.45 MHz (at −1.5 V, at an active area of 10 mm^2^; this value improved to 0.61 MHz at a smaller active area of 4 mm^2^). The rise and fall times of this OPD were rapid: 114 and 110 ns, respectively. Notably, the response speeds of solution-processed bulk-heterojunction (BHJ) OPDs are generally relatively slow (in the range 10–100 kHz), with only a few reported in the MHz region [32]. Accordingly, the performance of our solution-processed OPD is among the best ever reported, and suggests the possibility of developing optoelectronic devices incorporating OPDs prepared through such NFA ternary strategies.

## 2. Materials and Methods

The fabrication and characterization of the devices were described in Appendix A.

The preparation of active layer was described as follows. Poly((2,6-(4,8-bis(5-(2-ethylhexyl-3-fluoro) thiophen-2-yl)-benzo(1,2-b:4,5-b’]dithiophene))-alt-(5,5-(1′,3′-di-2-thienyl-5′,7′-bis(2-ethylhexyl)benzo(1′,2′-c:4′,5′-c’)dithiophene-4,8-dione)) (PBDB-TF-2F) (PM6, Organtec. Ltd., Chicago, IL, USA) were used as the donor (**denoted as D**), 2,2′-((12,13-Bis(2-butyloctyl)-12,13-dihydro- 3,9-dinonylbisthieno-(2′’,3′’:4′,5′)thieno-(2′,3‘:4,5)pyrrolo-(3,2-e:2′,3′-g)(1-3)benzothiadiazole-2,10-diyl)bis(methylidyne(5,6-chloro-3-oxo-1H-indene-2,1(3H)-diylidene)))bis(propanedinitrile) (BTP-eC9) (C9, Organtec. Ltd., Chicago, IL, USA) were used as the acceptor (**denoted as A1**) and 1-(3-methoxycarbonyl)-propyl-1-phenyl-(6,6)-C-71, (PC_71_BM, 1-Material Inc., Dorval, QC, Canada) was used as the dopant (**denoted as A2**). The donor, acceptor, and dopant were mixed at the weight ratio of D: A1: A2 = 1: 1.2: x (x = 0.1, 0.5, 1.0) in anhydrous chlorobenzene (CB) (99.8%, Sigma-Aldrich Inc., St. Louis, MO, USA). The mixture was stirred at 60 °C for 19 h to obtain the coating solution of the active layer. The prepared coating solution needs to be heated to 120 °C for 20 min before the coating process.

## 3. Results and Discussion

Figure 1a displays the materials used in this study. The center of BTP-eC9 features a fused-ring structure that can undergo efficient packing for enhanced carrier transport, and a side branch with a long carbon chain to ensure high solubility [17,23,33]. The energy levels of the lowest unoccupied molecular orbital (LUMO) and HOMO of the PM6 donor are −3.61 and −5.51 eV, respectively; those of the BTP-eC9 acceptor are −4.05 and −5.64 eV, respectively (Figure 1b). PM6:BTP-eC9–based OPVs have displayed PCEs of close to 18%, with good photocurrents, open-circuit voltages (*V*_OC_), and fill factors, making them good candidates for use in OPD applications [17]. The HOMO energy level of PC_71_BM ensures a large injection barrier with respect to PM6 or BTP-eC9; therefore, we excepted the system to suppress the injection dark current under reverse bias [34]. Previous reports have revealed that intermolecular interactions between PC_71_BM and NFA moieties can disrupt the formation of large NFA crystals and lead to compatible domains dispersed well within the conjugated polymer matrix. With a gradient distribution of the donor and acceptor moieties in a BHJ morphology, the presence of PC_71_BM can further suppress the dark current, while also providing efficient electron traps, to improve the performance of OPDs [5,35,36,37]. The embedding of PC_71_BM can be used to tailor the hole and electron mobilities of ternary blends and, thereby, balance charge transport [38,39,40].

To begin this study, we measured the contact angles, surface energies, and Flory–Huggins parameters (to determine the miscibility of the blend moieties) of the materials and their blend films. To provide the contact angles, we used water and diiodomethane as probe solvents. Appendix A lists the various parameters. Surface energies can be used as indicators of compatibility and to evaluate the distribution of the blend moieties. The interaction parameter χ is a measure of the compatibility and kinetic stability of a blend film morphology [41,42,43]. A low value of χ implies strong molecular interactions between the materials, leading to a well-mixed blend morphology. The values of χ for PM6 paired with BTP-eC9 and PC_71_BM were 0.35 and 1.07, respectively. The moderate value of χ for the PM6: BTP-eC9 pair suggested phase segregation of the blend. A high content of PC_71_BM can result in high degrees of phase segregation for PM6:PC_71_BM binary blends and also within ternary blends. The value of χ for BTP-eC9:PC_71_BM (0.19) was smaller than those of PM6: BTP-eC9 and PM6:PC_71_BM, suggesting that the interaction of PC_71_BM with BTP-eC9 was strong and would lead to their relatively good mixing.

We fabricated the photodiode OPDs to have the structure indium tin oxide (ITO)/ZnO/active layer (for a normal device: active area = 10 mm^2^)/MoO_3_/Ag (Figure 1c). We evaluated the performance of PM6:BTP-eC9 OPDs exhibiting an optimized blend ratio of 1:1.2 (control device), similar to that of optimized OPV devices [17,33,44,45]. We monitored the dark current of the control OPDs upon changing the thickness of the active layer. Appendix A reveals a decrease in the value of *J*_d_ of the OPDs upon increasing the thickness of the active layer from 100 to 160 nm; thereafter, it remained at a similar level for thicknesses up to 330 nm. Further increasing the thickness to 400 nm resulted in an increase in the dark current of the device. Therefore, we chose a thickness of 330 nm for the active layer of the normal binary OPD and for comparison with the ternary OPDs. Typically, a thicker active layer should lead to a device displaying a lower dark current; we attribute the higher values of *J*_d_ of the PM6: BTP-eC9 OPDs featuring thicker films to the high crystallinity of BTP-eC9. The ternary OPDs featured various blend ratios of the PM6 donor to the acceptors, from 1:1.2 to 1:2.2; we screened the feed ratios of BTP-eC9 and PC_71_BM to further optimize the conditions. Herein, we denote the OPDs incorporating PM6: BTP-eC9:PC_71_BM (1:1.2:X) ternary blends as “X–PCBM” devices; for example, the OPD featuring PM6: BTP-eC9:PC_71_BM at a weight ratio of 1:1.2:0.1 is denoted as the 0.1–PCBM device. The thickness of the OPDs was controlled to approximately 330 nm by adjusting the spin rate (Appendix A). Figure 2a displays the dark current behavior of the various OPDs. We observed a shift in the zero current voltage for the control and 1–PCBM OPDs, presumably because of internal charge accumulation during the dynamic processes of carrier injection, recombination, and transport [46]. The control PM6:BTP-eC9 displayed an excellent dark current at zero bias, but it underwent a rapid increase upon increasing the reverse bias. In general, optimizing the acceptor results in an OPV displaying an excellent value of *V*_OC_ (by minimizing the difference between the HOMO and LUMO energy levels of the donor and acceptor) and an optimized degree of light-harvesting (with a small band gap and acting as an NIR absorber). State-of-the-art NFA materials have a HOMO energy level (e.g., for Y6: −5.64 eV) close to that of the donor (e.g., for PM6: −5.51 eV), resulting in relatively low efficiency at blocking the hole injection current. The values of *J*_d_ (at a reverse bias of −1 V) *J_d_* of the control, 0.1–PCBM, 0.5–PCBM, and 1–PCBM devices were 1.68 × 10^−7^, 5.53 × 10^−7^, 3.27 × 10^−9^, and 1.78 × 10^−10^ A cm^−2^, respectively. Thus, significant decreases in the values of *J*_d_ occurred for the OPDs containing higher contents of PC_71_BM. This observation implies that a sufficient content of PC_71_BM enhanced the energy level barrier for inhibiting the injection current.

Next, we used optical microscopy (OM) and tapping-mode atomic force microscopy (AFM) to examine the blend morphologies and determine how they governed the OPD performance. Figure 3a presents OM images of the blend films. The control, 0.1–PCBM, and 0.5–PCBM films were featureless. A large degree of aggregation was evident for the film of 1–PCBM, presumably arising from the aggregation of PCBM. The surface roughness (Ra, under 1 × 1 µm scale) of the control, 0.1–PCBM, 0.5–PCBM, and 1–PCBM blends were 3.2, 3.8, 6.4, and 4.4 nm, respectively. Thus, the roughness of the 0.1–PCBM and 0.5–PCBM blends increased as a result of the presence of PCBM. Notably, the AFM image of 1–PCBM displayed no large-scale aggregation of PCBM, suggesting that the low value of Ra might have been due to the actual loading of PCBM being lower than that in 0.5–PCBM over the examined area. Appendix A presents AFM phase images of our blend films. We attribute the rod-like phase morphologies of these blends to highly crystalline BTP-eC9 domains. High densities of rod-like structures appeared for the control and 0.1–PCBM blends. Further increasing the content of PCBM led to a lower degree of rod-like structures, implying that PCBM inhibited the crystallinity of BTP-eC9. According to these morphological studies, one reason for lower dark currents for the 0.5–PCBM and 1–PCBM OPDs might have been their higher degrees of aggregation of PCBM, forming quenching sites that prevented the current from flowing out. Again, high loading of PCBM in the ternary blend provided an efficient injection barrier that suppressed the value of *J*_d_ of the OPD.

We used the two-dimensional fast Fourier transform (2D-FFT) model to examine the AFM images of the blend films, to better understand the respective directivity and symmetry, according to their phase diagrams [13,47,48]. To decompose the harmonic components of the AFM signals, we employed Gwyddion software for image analysis by applying the series of data of arbitrary dimensions without resampling; we applied filtering to exclude potential linear asymmetry, according to previous reports [13,48,49,50]. Figure 3c reveals that all of the films blended with PC_71_BM had the same diffraction peaks as those of PC_71_BM itself, indicating that the directionality and symmetry of the films improved after blending with PC_71_BM. The higher molecular order of the blend films would enhance their optoelectronic properties.

We performed grazing-incidence wide-angle X-ray scattering (GIWAXS) of the films of the blends and their pure materials to obtain information regarding their molecular packing. Appendix A reveals that the characteristic peaks of the films of pure PM6, BTP-eC9, and PC_71_BM appeared at 0.334, 1.624, and 1.348 Å^−1^, respectively. The positions of these characteristic peaks are consistent with those reported previously in the literature [17,45,51]. Two distinct patterns, located at 1.627 Å^−1^ (mainly contributed by the π-stacking of BTP-eC9) and 0.326 Å^−1^, appeared for the PM6:C9 blend film (Figure 4a). A weaker signal appeared at 1.6244 Å^−1^ for the 0.1–PCBM sample, indicating that the π-stacking of BTP-eC9 was weakened in the presence of PCBM (Figure 4b). Further increasing the PCBM content (i.e., 0.5–PCBM) led to the disappearance of the signal at 1.627 Å^−1^ (Figure 4c), suggesting that the embedding of a high content of PCBM inhibited the crystallization of BTP-eC9, consistent with the findings from the AFM phase images. For the 1–PCBM sample, the signal at 1.627 Å^−1^ reappeared, along with a signal at 1.3187 Å^−1^ representing crystalline PCBM (Figure 4d), implying that over-aggregation of PCBM allowed BTP-eC9 to recover its crystallinity. A higher content of PCBM led to large degrees of aggregation and/or crystal formation in the 1–PCBM sample (consistent with its OM image). Figure 4e displays cartoon images of the blend film morphology; the differences in miscibility among PM6, BTP-eC9, and PCBM were responsible for the variations in the morphologies of the ternary blends, with PCBM suppressing the molecular packing of BTP-eC9. These changes in morphology (aggregation of PCBM; π-stacking of BTP-C9) for 0.5–PCBM and 1–PCBM were responsible for their lower values of *J*_d_ [29].

We determined the LDRs of our OPDs under various light intensities. To avoid overestimating their values, we calculated the LDRs (without considering the noise current) by using the following Equation (1) [52]:(1)$$\mathrm{LDR}\;\left(\mathrm{dB}\right)=20\times \log\left(\frac{J_{\max}}{J_{\min}}\right)\;$$
where *J*_max_ is the current feedback under the strongest light intensity and *J*_min_ is the current generated under low light intensity (greater than the dark current). At 0 V, the LDRs of the control, 0.1–PCBM, 0.5–PCBM, and 1–PCBM OPDs were 106.2, 139.4, 142.4, and 144.6 dB, respectively. Figure 2c reveals that, under a −1 V bias, the LDRs of the 0.5–PCBM and 1–PCBM OPDs were 102.3 and 106.3 dB, respectively. Because it lowered the value of *J*_d_, the embedding of PCBM extended the LDR of the OPDs.

We measured the external quantum efficiencies (EQEs) and responsivities (R) of the OPDs to evaluate the effects of the blend ratios and morphologies. The EQE of the PM6: BTP-C9 binary OPD was close to 70% under zero bias (Figure 5a). The EQE responses of the 0.1–PCBM and 0.5–PCBM OPDs were similar, but that of the 1–PCBM device was lower, suggesting that over-aggregation of PCBM increased the content of carrier traps, which inhibited the extraction of free carriers. Minimizing the dark current while maintaining high responsivity is a prerequisite for a useful OPD. The photocurrent and dark current are competitive, with an increase in the bulk resistance of the blend limiting both the current extraction-out and injection-in. As a result, the lowest value of *J*_d_ was that of the 1–PCBM OPD, but it occurred while sacrificing its EQE response. The EQE curves of the 0.1–PCBM and 0.5–PCBM OPDs revealed EQE gains of greater than 10% at a reverse bias of −1.5 V, with the highest response at 860 nm of up to 100%. Notably, we could not obtain reasonable EQE responses from the control and 1–PCBM OPDs, due to a series carrier injection phenomenon (Appendix A; we checked this phenomenon several times for accuracy). We suspected that the high molecular order (as revealed in the AFM phase images and GIWAXS patterns) and relatively high HOMO energy level of BTP-eC9 increased the injection current under reverse bias, leading to distortion of the EQE responses of the control OPDs; over-aggregation of PCBM in the 1–PCBM OPD led to a similar phenomenon. We calculated the values of R according to Equation (2) [53]:(2)$$R\left(\lambda \right)=\mathrm{EQE}\left(\lambda \right)\times \frac{e}{h\upsilon }\left(A\;W^{-1}\right)\;$$
where e is the elementary charge, and hυ is the incident photon energy. Figure 5b presents plots of R measured at biases of 0 and −1.5 V. At 0 V, the values of R at λ_max_ (860 nm: R_λmax_) of the control, 0.1–PCBM, 0.5–PCBM, and 1–PCBM OPDs were 0.43, 0.45, 0.42, and 0.41 A W^−1^, respectively. At −1.5 V, the values of R_λmax_ of the 0.1–PCBM and 0.5–PCBM OPDs were 0.572 and 0.59 A W^−1^, respectively. To the best of our knowledge, this value of R for the 0.5–PCBM OPD is among the highest ever reported for a broadband NIR OPD. We used Equation (3) to calculate the detection ratio [6,30]:(3)$$D^{*}=\frac{R}{\sqrt{2q\;J_{\mathrm{dark}}}}\;\left(\mathrm{Jones}\right)$$
where *D** is the detectivity (Jones), *q* is the basic charge (1.6 × 10^–19^ C), and *J*_dark_ is the dark current density, which is dominated by shot noise when ignoring the Johnson noise and Flicker noise (1 f^−1^). We recorded the plots of *D** under 0 and −1.5 V bias. At 0 V, the control, 0.1–PCBM, and 0.5–PCBM OPDs provided values of *D** of greater than 1 × 10^14^ Jones at the wavelengths between 350 and 850 nm (Appendix A), because of the same order of magnitude of their values of *J*_d_ (at 0 V, Figure 2a) and their similar responsivities (Figure 5a). Again, we could not determine the values of *D** for the control and 1–PCBM OPDs under reverse bias because of the unreasonable values of *R*. Figure 5c reveals the best device that the value of *D** was greater than 2 × 10^12^ Jones (at −1.5 V), with a maximum value of 1.1 × 10^13^ Jones at 830 nm. Appendix A compares our values of *D** with those described previously in the literature; our reported values represent the best performance among the broadband OPDs [2,10,18,19,20,22,29,31,32,54,55,56,57,58,59,60]. Thus, appropriate blending of PC_71_BM can improve the dark current performance and detectivity of OPD devices.

We determined the rise and fall times (for amplitudes between 90 and 10%) by measuring the transient photovoltage (at 530 nm with a light source frequency of 1 kHz under –1 V) of the OPDs, to evaluate the accumulation of carriers. The corresponding maximum available frequencies (cut-off frequency at −3 dB: *f*_−3dB_) of the OPDs revealed the applicable bandwidth [13]. Figure 6 summarizes the results. The rise/fall times of the control, 0.1–PCBM, 0.5–PCBM, and 1–PCBM OPDs were 52.6/60.1, 2.5/5.6, 2.3/1.8, and 4.5/1.6 μs, respectively. The control OPD had relatively lower response speeds, with the on/off switching of the optical signal displaying distortion in the falling curve region, implying a slow transfer of the charge. In contrast, the embedding of PCBM allowed the OPDs to reach the steady-state photocurrent and dark current without distortion of the waveform between the rise and fall signal (Figure 6a). The 0.5–PCBM device displayed a fast on/off response (over one order of magnitude faster than the control OPD) because its optimized blend morphology contributed to balanced charge carrier transport. The efficient charge separation and charge transfer resulting from the retrained trap density of the OPDs in the presence of PCBM promoted their response times [2]. We used space charge limited current (SCLC) measurements to determine the hole and electron mobilities of the OPDs. We fabricated hole- and electron-only devices to have the structures ITO/PEDOT: PSS/active layer/MoO_3_/Au and ITO/ZnO/active layer/Ca/Al, respectively. The electron/hole mobilities of the control, 0.1–PCBM, 0.5–PCBM, and 1–PCBM OPDs were 2.3/9.7, 3.4/9.5, 1.1/4.6, and 9.5/5.5 × 10^−4^ cm^2^ V^−1^ s, respectively. The difference between the hole and electron mobilities was lowest for the 0.5–PCBM OPD, confirming that balanced carrier transport contributed to its rapid response time. We also determined the rise and fall times of a 0.5–PCBM device with a device area of 0.04 cm^2^ (Appendix A). The rise and fall times of this smaller device were 114 and 100 ns, respectively; the smaller active layer area led to fewer defects in the blend film and allowed the OPD to exhibit response times on the order of nanoseconds. The small-area device provided a dark current of 3.27 × 10^−9^ A cm^−2^ at −1 V (Appendix A). In addition, we determined the optoelectronic properties of a commercially available Si-PD (S1336-44BQ, Hamamatsu, Japan) designed for precision photometry from the UV to the NIR region. Appendix A reveals that the values of *D** for our devices were comparable with those of the Si-PD. The determined rise and fall times of the Si-PD were 588 and 584, respectively. The response times of our 0.5–PCBM device were not only among the fastest for broadband OPDs but also outperformed those of the Si-based PD detector (Appendix A). Figure 6b displays the cut-off frequency curves of our OPDs. The values of *f*_−3dB_ of the control, 0.1–PCBM, 0.5–PCBM, and 1–PCBM OPDs were 15, 200, 450, and 420 kHz, respectively. The values of *f*_−3dB_ of solution-processed BHJ OPDs are typically in the range from 10 to 100 kHz, with only a few reports of values in the MHz region [32]. Again, the cut-off frequency of our 0.5–PCBM OPD (0.45 MHz) is comparable with those of most other previously reported OPDs. Furthermore, the value of *f*_−3dB_ of the 0.5–PCBM OPD improved to 0.61 MHz when the device had an active area of 0.04 cm^2^; in comparison, the value of *f*_−3dB_ of the Si-PD (S1336-44BQ, Hamamatsu, Japan) was 0.5 MHz. The 0.5–PCBM device displayed an extremely low dark current of less than 10^−9^ A cm^−2^ under reverse bias and a high value of *R**_λ_*_max_ of 0.59 A W^−1^ in the NIR region, along with values of *D** of up to 1.1 × 10^13^ Jones. The cut-off frequency and response time of our 0.5–PCBM device were better than those of the commercial Si-PD, highlighting its potential for use in applications requiring rapid responses (e.g., image photosensors; medical monitoring) [29]. Again, this performance is the best ever reported for a broadband NIR OPD.

## 4. Conclusions

Blending of PC_71_BM as the third component in PM6: BTP-eC9 OPDs effectively suppressed the dark currents of the devices. The deep HOMO energy level of PC_71_BM, and its effect on altering the blend film morphology, resulted in efficient blocking of the carrier injection under reverse bias. The optimized 0.5–PCBM OPD displayed high responsivity of 0.59 A W^−1^ at 860 nm (at −1.5 V) with an ultralow value of *J*_d_. At biases of 0 and −1.5 V, we measured high OPD detection ratios of 3.4 × 10^14^ and 1.1 × 10^13^ Jones, respectively. We measured a value of *f*_−3dB_ of 0.61 MHz for this device, associated with rise and fall times of 114 and 100 ns, respectively. Thus, our 0.5–PCBM OPD exhibited state-of-the-art OPD performance, even outperforming a commercial UV–IR Si-PD (from the visible up to 860 nm). This new strategy for device fabrication is, therefore, a feasible approach for achieving highly efficient broadband OPDs.

## Figures and Tables

**Figure 1 nanomaterials-12-01378-f001:**
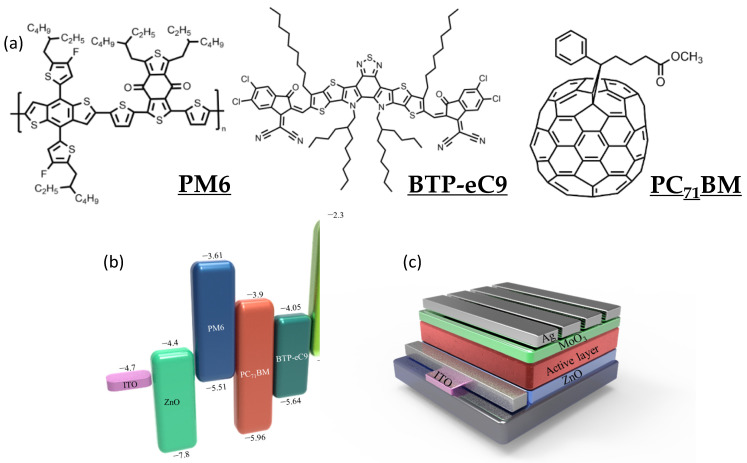
(**a**) Chemical structures and (**b**) energy levels of the materials used to prepare the OPDs. (**c**) Structure of the OPDs.

**Figure 2 nanomaterials-12-01378-f002:**
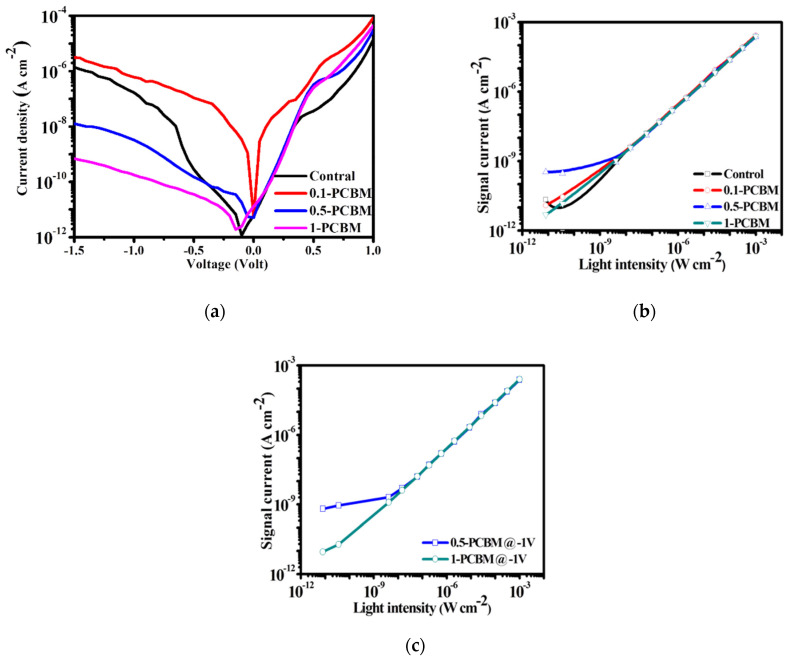
(**a**) Dark current curves of the OPDs. (**b**) LDRs of the OPDs under zero bias. (**c**) LDRs of 0.5–PCBM and 1–PCBM at −1 V.

**Figure 3 nanomaterials-12-01378-f003:**
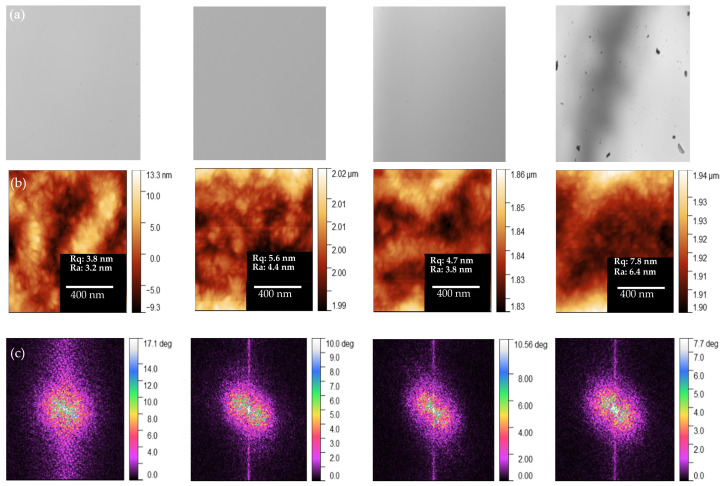
(**a**) OM images, (**b**) topographical and (**c**) 2D-FFT–converted AFM images of the blend films.

**Figure 4 nanomaterials-12-01378-f004:**
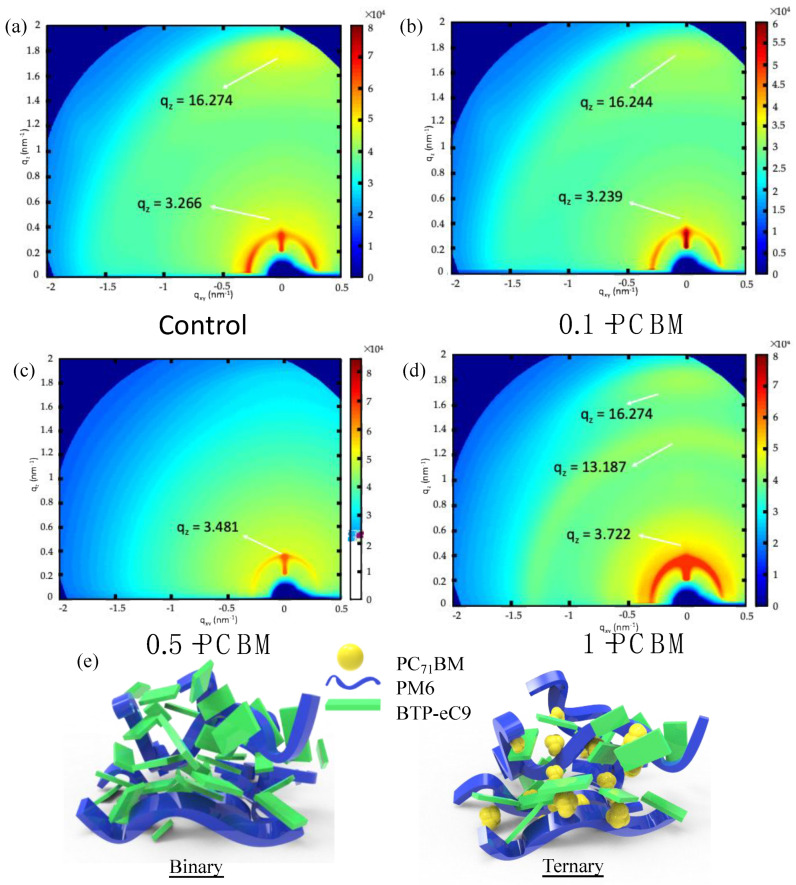
(**a**–**d**) GIWAXS images of the (**a**) binary, (**b**) 0.1–PCBM, (**c**) 0.5–PCBM, and (**d**) 1–PCBM blends. (**e**) Cartoon illustration of blend morphology.

**Figure 5 nanomaterials-12-01378-f005:**
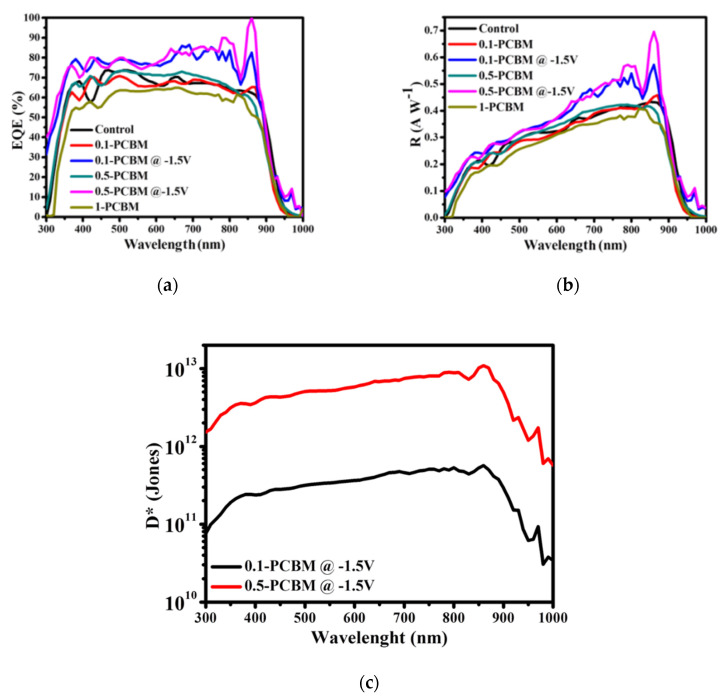
(**a**) EQE responses and (**b**) responsivity curves of the OPDs at biases of 0 and −1.5 V. (**c**) Plots of *D** for the 0.1–PCBM and 0.5–PCBM OPDs at a bias of −1.5 V.

**Figure 6 nanomaterials-12-01378-f006:**
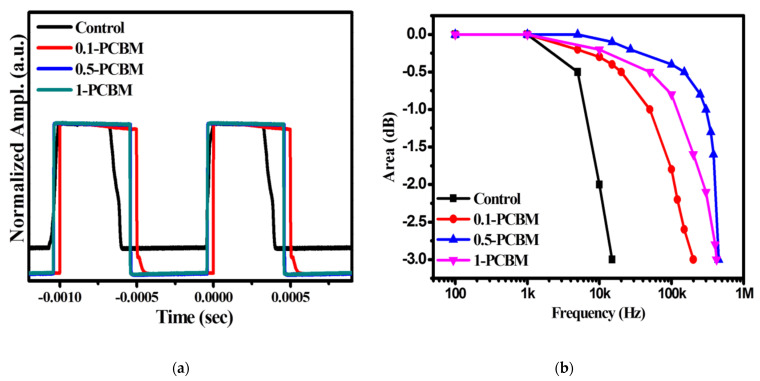
(**a**) Rise and fall times and (**b**) cut− off frequencies of the OPDs.

## Data Availability

Data are contained within the article or Appendix A.

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
