# Peer review of "Realizing Broadband NIR Photodetection and Ultrahigh Responsivity with Ternary Blend Organic Photodetector"

_nanomaterials, 2022, doi:10.3390/nano12081378_

Round 1

Reviewer 1 Report

The authors present an experimental realization of near-infrared organic photodiodes (OPDs) having a broad bandwidth (350–950 nm), ultrahigh responsivity, and a high photoresponse frequency. For this purpose, they used a ternary blend strategy to prepare PM6:BTP-eC9:PCBM–based OPDs displaying an extremely low dark current  (~10-9A cm-2) and high detectivity (~1013 Jones) at 860 nm. The developed devices outperform commercial inorganic Si photodetectors. I found that the idea and approach not very original, although its physical realization worths recognition. The manuscript can be published in Nanomaterials.

Author Response

Thank you very much for the referees’ comments on our manuscript.

Reviewer 2 Report

The topic of OPDs is a very important nowdays, which makes the paper actual and important. The presented results are novel, and the conducted experiments support the conclusions made. This can suffice to make the paper suitable for publication, if not the awfully careless preparation. Please revise the whole paper from this point of view! I present below just several examples, which are, however, much more numerous, and they make reading the paper an unpleasant reading.

Here are some examples:  

1) Please check for misprints, such as "(114 and 110 ns, respectively.." in the abstract (no closing bracket + double fullstop).

2) Figures 2, 4 (at least) brakes the sentence

3) Fig 3 caption includes double "and"

4) "Å –1", "W–1", etc. should contain a superscript

 5) Lines 282-293: does the Eq. (3) correspond to the text or to the Fig. 5? The text size is that of the Fig caption, and the Figures brakes the text. 

Author Response

Thank you very much for the referees’ comments on our manuscript. We have carefully modified our manuscript accordingly. We have highlighted all the changes by giving the text a yellow background in marked copy of the manuscript.
